# Plasma Circular RNAs as Biomarkers for Breast Cancer

**DOI:** 10.3390/biomedicines12040875

**Published:** 2024-04-16

**Authors:** Domenico Tierno, Gabriele Grassi, Fabrizio Zanconati, Barbara Dapas, Bruna Scaggiante

**Affiliations:** 1Department of Medicine, Surgery and Health Sciences, University of Trieste, Strada di Fiume 447, I-34149 Trieste, Italy; tiernodomenico@gmail.com (D.T.); ggrassi@units.it (G.G.); fabrizio.zanconati@asugi.sanita.fvg.it (F.Z.); 2Department of Chemical and Pharmaceutical Sciences, University of Trieste, Via L. Giorgieri 1, I-34127 Trieste, Italy; bdapas@units.it; 3Department of Life Sciences, University of Trieste, Via Valerio 28, I-34127 Trieste, Italy

**Keywords:** breast cancer, liquid biopsy, circRNA, biomarker, circulating circRNA, plasma, extracellular vesicles

## Abstract

Breast cancer (BC) is currently the most common neoplasm, the second leading cause of cancer death in women worldwide, and is a major health problem. The discovery of new biomarkers is crucial to improve our knowledge of breast cancer and strengthen our clinical approaches to diagnosis, prognosis, and follow-up. In recent decades, there has been increasing interest in circulating RNA (circRNA) as modulators of gene expression involved in tumor development and progression. The study of circulating circRNAs (ccircRNAs) in plasma may provide new non-invasive diagnostic, prognostic, and predictive biomarkers for BC. This review describes the latest findings on BC-associated ccircRNAs in plasma and their clinical utility. Several ccircRNAs in plasma have shown great potential as BC biomarkers, especially from a diagnostic point of view. Mechanistically, most of the reported BC-associated ccircRNAs are involved in the regulation of cell survival, proliferation, and invasion, mainly via MAPK/AKT signaling pathways. However, the study of circRNAs is a relatively new area of research, and a larger number of studies will be crucial to confirm their potential as plasma biomarkers and to understand their involvement in BC.

## 1. Breast Cancer

Breast cancer (BC) is the most diagnosed cancer in women worldwide, with 297,790 new cases in 2023 (31% of all cancer diagnosis in women), and the second leading cause of tumor-related of death in women with 43,170 new cases in 2023 (15% of all tumor-related deaths in women) [1]. For this reason, BC has a major social impact worldwide; the most important goal remains to reduce the risk factors and detect the disease at an early stage to ensure treatment with the best chance of cure. The challenge is not only to research therapeutic treatments, but also to develop novel biomarkers useful for screening programs, disease prognosis and clinical follow-up of patients. New risk prediction models need to be developed to stratify screening options and improve cost-effectiveness, and in this regard, new biomarkers could help achieve the goal. Similarly, clinical research needs to discover innovative biomarkers to predict tumor progression and response to treatment, which could lead to effective therapeutic targets. From this point of view, liquid biopsies represent an attractive tool, as they can be easily standardized in routine clinical practice [2,3]. In recent years, the use of liquid biopsies has become particularly interesting for the detection of circular RNAs (circRNAs), a class of non-coding RNAs (ncRNAs) that are attracting increasing attention as modulators of gene expression through interaction with DNA, miRNAs, and proteins. The findings on the function of circRNAs point to a new hierarchical level in biological process regulation, and emphasize the need for their in-depth investigation. Indeed, dysregulation of circRNA expression is often associated with the onset and progression of various human diseases, including cancer. Accordingly, circRNAs may represent novel and effective diagnostic, prognostic, and predictive biomarkers for human diseases. In addition, the higher stability compared to linear RNAs makes circRNAs excellent candidates for non-invasive biomarkers in liquid biopsy. In the following sections, we will discuss the most interesting evidence for the potential role of circulating circRNAs (ccircRNAs) in plasma as BC biomarkers.

## 2. circRNAs

The covalently closed circular RNAs (circRNAs) are a class of non-polyadenylated RNAs that were originally identified in plant viroids, viruses, and yeast [4]. They arise as secondary products of non-canonical splicing of pre-mRNA, and were therefore considered for many years as intermediate molecules of no biological relevance [5]. Nevertheless, the improvement of enrichment techniques for non-polyadenylated RNAs coupled with high-throughput RNA sequencing (RNA-seq) has identified a broad class of circRNAs with tissue-specific expression, suggesting an active biological role for these non-coding RNAs [6,7]. The involvement of circRNAs in biological processes and human diseases has become increasingly attractive. A growing number of studies suggest that circRNAs play a crucial role in the regulation of biological processes at the DNA, RNA, and protein levels [8]. Indeed, their dysregulation is frequently observed in the development and progression of human diseases [9]. Although their biology remains to be further elucidated, their high stability, abundance, and specific expression suggest a relevant clinical potential as novel biomarkers or therapeutic targets.

### 2.1. Biosynthesis

circRNAs generally arise from the circularization of exons in pre-mRNA by back-splicing, a non-canonical splicing event that differs from canonical splicing sites by the splice sites involved. While in canonical splicing, a 5′ upstream donor splice site is joined to a 3′ downstream acceptor site (Figure 1A), in back-splicing, a downstream donor splice site is reversibly joined to an upstream acceptor site (Figure 1B). This splicing event leads to a conformational change in the pre-mRNA, resulting in the skipping of exons between the joined splice sites and their circularization by the spliceosome machinery. Back-splicing thus generates a covalently closed circRNA, and a linear RNA with the remaining exon that is rapidly degraded. Notably, both mechanisms are mediated by the splicing machinery [10].

Homeostasis between canonical splicing and back-splicing is regulated by various endogenous processes, such as RNA pairing between reverse complementary sequences in pre-mRNA introns. For example, RNA pairing in different introns brings the downstream donor splice site close to the upstream acceptor site, which promotes black-splicing. Repetitive sequences such as ALU or long complementary sequences in different introns favor this pairing (Figure 2A) [11,12]. Instead, RNA pairing between complementary sequences within a single long intron promotes canonical splicing and exon inclusion by bringing the upstream donor site closer to the downstream acceptor site for the spliceosome machinery [13]. Similar to the RNA pairing of introns, proteins that bind pre-mRNA can trigger back-splicing in addition to the canonical splicing event (Figure 2B). This is the case for MBL, a dimeric protein that binds the exon 2 flanking introns of its own pre-mRNA to bring the downstream donor splice site and the upstream acceptor site for back-splicing closer [14]. It is important to emphasize that circRNA synthesis is highly stratified and can undergo several alternative circularizations. For example, multiple circRNAs containing a specific set of exons can be generated from a single pre-mRNA by competition of several ALUs or complementary sequences across different introns. Similarly, a single pre-mRNA can generate multiple exon circRNAs that differ by the inclusion or absence of interspaced introns [6,12]. The regulatory mechanisms of intron retention in exon circRNAs are still unclear.

CircRNA assembly is not limited to back-splicing, but can also involve introns by stabilizing the intron lariat that has escaped from debranching after splicing. Intron circRNAs are generated by consensus sequences near the 5′ splice site and the branch-point site, which are paired to circularize the intron element (Figure 2C) [15]. The synthesis of an intron or exon circRNA is not mutually exclusive, as a single pre-mRNA can give rise to both circRNAs, through back-splicing, and the stabilization of the intron lariat. Finally, recent studies have shown that circRNAs can arise not only from pre-mRNA, but also from other RNAs, e.g., long non-coding RNAs (lncRNA) and antisense transcripts [16]. These findings indicate the complexity of circRNA synthesis, and point to the need for in-depth investigation on their biogenesis.

### 2.2. Biological Function

A complete understanding of the biological role of circRNAs is still being explored. So far, the literature suggests that most circRNAs act as miRNA sponges to regulate post-transcriptional gene expression (Figure 3A) [17,18]. Since they act as endogenous competitors of miRNA–mRNA binding sites, it is not surprising to observe a high abundance of competing binding sites for miRNA in several circRNAs [19]. Dysregulation of circRNAs is frequently associated with the onset and progression of numerous human diseases.

The dysregulation of circRNAs acting as miRNA-sponges can influence several pathological events. For example, overexpression of circCD44 promotes the tumorigenesis of triple-negative breast cancer by sponging miR-502-5p, a miRNA involved in the regulation of the RAS pathway [20]. Instead, sponging of miR-145 (a miRNA involved in cell motility) by circLRP6 promotes the progression of atherosclerosis [21]. It is important to note that circRNAs can sponge multiple miRNAs, leading to a complex network regulation that may be critical for pathological effects if dysregulated. For example, upregulation of the single-stranded circRNA CDR1as promotes tumor progression of osteosarcoma [22], colorectal carcinoma [23,24], lung carcinoma [25], pancreatic ductal adenocarcinoma [26] by sponging miR-7, hepatocellular carcinoma by sponging miR-1285 [27], and triple-negative breast cancer by sponging miR-1299 [28]. In contrast, CDR1as inhibits the progression of bladder carcinoma by sponging miR-135a [29]. Moreover, CDR1as was found to be under-expressed in the brain of Alzheimer’s patients, suggesting that it plays a role in the development of the disease via miR-7 regulation [30].

The regulatory ability of circRNAs is also evident in their interaction with circRNA binding proteins (cRBPs). Due to their capillary regulation of protein interactions, cRBPs play an essential role in several pathogenic conditions. The functional significance of this interaction can be manifold: recruitment of proteins to specific cellular sites and regulation of protein–protein, protein–RNA, and protein–DNA interactions (Figure 3B) [31]. A classic example is circ-FOXO3, a cRBP that interacts with MDM2 and p53 to catalyze p53 ubiquitination and proteasome-mediated degradation [32]. Circ-FOXO3 binds and stabilizes the CDK2/p21 complex to inhibit the formation of the CDK2/cyclin E complex and arrest the cell cycle in G1 phase [33].

However, cRBPS can also interfere with the interactions of proteins with DNA, RNA, or other proteins [34]. This is the case with cia-cGAS, the circRNA antagonist of cyclic GMP-AMP synthase (cGAS), a DNA-binding protein involved in the response to microbial infection. The binding of cGAS to DNA stimulates the synthesis of cAMP-cGMP which, in turn, triggers the synthesis of type I interferon via the STING pathway. To avoid auto-immune events, cia-cGAS sequesters cGAS and blocks its binding to DNA under homeostatic conditions [35,36]. Instead, circANRIL binds and occupies the protein PES1, a nucleolar pre-ribosomal assembly factor, to hinder the pre-rRNA maturation and ribosome assembly. This leads to nucleolar stress and activation of apoptosis via p53, especially in atherosclerotic plaques [37]. An example of cRBP inhibition of protein–protein interaction is circCCNB1, which binds to both CCNB1 and CDK1, thus blocking complex formation and cell cycle progression [38].

Circular RNAs cannot encode proteins via canonical cap-dependent translation because they lack a 5′ cap and a polyadenylated 3′ tail. Nevertheless, recent studies have identified several circRNAs that encode proteins through two cap-independent translations: the first is via the internal ribosome entry sites (IRESs), the second is via the incorporation of N6-Methyladenosine (m6A) into the 5′ untranslated region (5′-UTR). Both IRESs and m6A recruit ribosomes and translation initiation factors to circRNAs, replacing the role of the 5′ cap [39,40] (Figure 3C). The peptides encoded by circRNAs are essentially truncated versions of the original protein that can have independent, synergistic, or opposing functions. For example, SHRPH-146aa, encoded by circ-SHRPH, protects the full-length protein SHRPH, an onco-suppressor involved in DNA repair, by acting as a decoy for ubiquitination, to prevent degradation by the proteasome [41].

The synthesis of circRNAs can be viewed as a translational regulatory process, since back-splicing and canonical splicing are largely mutually exclusive. This means that the assembly of a circRNA inhibits the formation of its mature host mRNA and, subsequently, translation (Figure 3D). For example, the amount of MBL protein in the cells regulates the formation of circMBL or mature MBL mRNA through a feedback loop. Basically, overexpression of MBL stimulates the formation of circMBL, while its low amount leads to canonical splicing for appropriate MBL mRNA maturation [14].

Increasing evidence for the importance of circRNAs in human disease emphasizes their potential as efficient biomarkers and novel therapeutic targets. One example is the study by Galardi et al. describing the crucial role of circRNAs in pediatric tumors [42]. Among the most promising circRNAs reported, circSKA3 plays a crucial role in promoting cell proliferation in medulloblastoma, one of the most common brain tumors in children, by sponging miR-383-5p, miR-326 and miR-520 h [43,44,45]. Similarly, hsa_circ_0000527 and hsa_circ_0000034 stimulate the proliferation and invasion of retinoblastoma tumor cells by sponging miR-646 and miR-361-3p, respectively [46,47,48]. Exon skipping is an interesting new therapeutic approach based on the removal of aberrant exons and the production of truncated proteins to partially restore the physiological cellular phenotype. Anti-sense circRNAs could be novel efficient tools to achieve exon skipping in target aberrant proteins [49]. In addition, the potential clinical applications of circRNAs have been extended to their use as novel nucleic acid vaccines, but many manufacturing and safety limitations still need to be overcome [50].

## 3. Liquid Biopsy and Circulating circRNAs

Liquid biopsy is the analysis of molecules from biological fluids of great clinical relevance, such as blood, urine, cerebrospinal fluid, ascites, and pleural fluid. Liquid biopsies from blood and urine are minimally invasive, and can be the source of various biomarkers useful for cancer surveillance [51]. Some of the better studied components of body fluids that are of clinical interest are:Circulating tumor cells (CTCs): As the name suggests, CTCs are cells that have detached from a tumor and migrated into the bloodstream. They have a low concentration (about 10 cells/mL blood), but can help to preserve the biological, molecular, and histological characteristics of the primary tumor [52,53].Cell-free circulating nucleic acids (ccfNAs): These are essentially nucleic acids, DNA and RNA, that circulate freely in body fluids. Like CTCs, they have a low concentration and are highly fragmented. Nevertheless, the concentration, presence of mutations and integrity of ccfNAs have been shown to provide important diagnostic, prognostic and predictive information in cancer [54].Tumor-educated platelets (TEPs): These are platelets that have been altered by the tumor through the exchange of biomolecules to make them receptive to cancer signals. They play an active role in metastasis by covering the tumor cells in the bloodstream and protecting them from the immune system. Therefore, the concentration and molecular composition of TEPs can provide useful information about tumor progression [55].Exosomes: These are a class of extracellular vesicles that transport biomolecules from the cells of origin. Exosomes can be found in the body fluids of patients with various diseases and, therefore, have great potential as biomarkers. The characterization of their content is helpful to deepen the molecular profile of the pathological cells of origin [56].

Circulating circRNAs (ccircRNAs) are nucleic acids with potential clinical attractiveness in liquid biopsies. Their usefulness as biomarkers is based on their remarkable abundance (more than 2400 ccircRNAs have been identified in human blood) and their higher stability compared to linear RNAs [4,57]. The clinical significance of ccircRNA in liquid biopsy is becoming increasingly clear in the literature. For example, lower expression of circKIAA1244 in the plasma of patients with gastric cancer is associated with an increased risk of metastasis and a poor outcome [58]. In addition to cell-free ccircRNAs, ccircRNAs packaged in exosomes also have a potential clinical benefits. High serum levels of exosomal circIARs in patients with pancreatic tumors are associated with metastasis, advanced tumor stage, and poor outcome. Indeed, this circRNA is involved in the sponging of miR-122 and ZO-1, a tight junction protein, as well as in the overexpression of RhoA and RhoA-GTP, crucial genes for tumor invasion and metastasis. In addition, circIARs was also overexpressed in pancreatic tumor tissue, indicating a positive molecular correlation between tissue and plasma, at least as far as this circRNA is concerned [59].

As knowledge of circRNAs and their biological function deepens, their potential as biomarkers is becoming increasingly clear. This review discusses the current state of knowledge of breast cancer-associated ccircRNAs in plasma and their utility as biomarkers. The articles were searched in PubMed using the keywords “circulating”, “circRNA”, “plasma”, and “breast cancer”, and dated from January 2017 to January 2024. Since each host gene can produce multiple circRNAs, the latter are indicated with their circBase ID (hsa_circ_xxxxxxx) to avoid confusion. A graphical abstract summarizing the key points and main findings of this review is reported in Figure 4.

## 4. BC-Associated ccircRNAs

Plasma investigation of ccircRNAs of clinical interest in BC is characterized by a recurrent workflow in the reviewed studies: (1) identification of the most dysregulated circRNAs in BC by microarray, NGS techniques, in silico analysis, and literature search; and (2) validation of selected circRNAs in plasma samples by qRT-PCR. Here, we discuss the key findings relevant to the understanding the importance of ccircRNAs as biomarkers and in the pathogenesis of BC.

### 4.1. Identification of ccircRNAs by Expression Profiling of Plasma

Yin et al. [60] first performed a microarray assay on five pairs of BC and healthy plasma samples, and found 49 significantly dysregulated ccircRNAs in BC samples (19 upregulated and 22 downregulated compared to controls). From the latter, Yin et al. [60] selected the three most dysregulated ccircRNAs (hsa_circ_0001785, hsa_circ_0108942, and hsa_circ_0068033) for validation in a second sample pool of 20 pairs of BC and healthy plasma samples by qRT-PCR. The results showed a significant overexpression of hsa_circ_0001785 and hsa_circ_0108942, and a downregulation of hsa_circ_0068003 expression in BC plasma compared to healthy controls. Since hsa_circ_0001785 had a higher diagnostic potential, Yin et al. [60] further analyzed it in 57 matched pre-operative and post-operative plasma from BC patients by qRT-PCR. The analysis showed a strong correlation between the plasma level of hsa_circ_0001785 and histological grade, TNM stage, and distant metastasis, but not with age, lymph node invasion, and hormone receptor status. Moreover, the plasma level of hsa_circ_0001785 decreased significantly after surgery, indicating a possible predictive value of this ccircRNA [60]. The authors did not perform a functional analysis of the identified circRNAs, but other studies delved into the role of hsa_circ_0001785, hsa_circ_0108942, and hsa_circ_0068033 in BC progression. Zhiyang Li et al. indicated that hsa_circ_0001785 inhibits cell proliferation and invasion in BC cell lines (T47D, and MDA-MB-231) through miR-942 sponging and subsequent upregulation of SOCS3 [61], an oncosuppressor involved in cytokine signaling [62]. This evidence appears to contradict the findings of Yin et al. [60], who associated the upregulation of hsa_circ_0001785 with the occurrence of BC and poorer clinical outcomes. In human primary tumors, which are highly complex, other molecular signaling pathways may also be involved. Further investigation into the functional role and mode of plasma secretion of hsa_circ_0001785 in BC is warranted. With regard to other ccircRNAs overexpressed in BC, Chuansheng Yang et al. have shown that upregulation of hsa_circ_0108942 in BC cell lines BT-549 and MDA-MB-468 stimulates cell proliferation and invasion by sponging miR-1178-3p and upregulating TMED3 [63], a protein involved in activation of the β-catenin/Wnt pathway [64]. Pengfei Yuan et al. found that upregulation of hsa_circ_0068003 in BC cell lines (MCF-7 and MDA-MB-231) resulted in inhibition of cell proliferation and promotion of apoptosis [65], thus acting as an onco-suppressor which may fit with the under-expression in plasma of BC patients reported by Yin et al. [60].

Lin et al. [66] performed a comprehensive characterization of ccircRNA expression profile of extracellular vesicles (EVs) in BC to develop a ccircRNA expression panel for diagnosis. First, they use Illumina NGS to characterize the expression profile of EVs long RNAs in the plasma of 14 BC and 6 Breast Benign Tumor (BBT) patients. The analysis revealed an overall percentage of ccircRNA in EVs of about 28% (the others were 56% for lncRNA and 16% for mRNA). The comparison between BC and BBT EVs showed a significantly higher ccircRNA level in BC compared to BBT and 439 dysregulated ccircRNAs (i.e., 162 overexpressed and 277 underexpressed in BC EVs). The 20 most-upregulated ccircRNAs in EVs in BC were then analyzed by qRT-PCR in EVs from a second cohort of 101 BC patients and 81 controls (30 healthy and 51 BBT). The results confirmed the upregulation of these ccircRNAs in EVs from BC patients compared to controls. Using multiple machine learning models, the authors were able to select a panel of nine ccircRNAs with the greatest ability to discriminate BC cases from healthy controls. The panel consisted of hsa_circ_0002190, hsa_circ_0007177, hsa_circ_0000642, hsa_circ_0001439, hsa_circ_0001417, hsa_circ_0005552, hsa_circ_0001073, hsa_circ_0000267, and hsa_circ_0006404. The diagnostic efficiency of this panel was evaluated by qRT-PCR in a third cohort of 43 BC patients and 34 controls (13 healthy and 21 BBT). Again, the results confirmed the great potential of this panel for BC diagnosis [66], and indicated the need for functional testing of the nine selected ccircRNAs.

Youting Hu et al. [67] characterized the expression profile of ccircRNAs in six BC and six healthy control plasma samples by microarray to assess dysregulated ccircRNAs with clinical relevance. Among the resulting deregulated ccircRNAs in BC, they found hsa_circ_0008673 as the most notable upregulated ccircRNA, which was selected for further analysis. To validate the ccircRNA, qRT-PCR quantification was performed on 378 preoperative BC and 102 healthy control plasma samples. The results confirmed a significant upregulation of hsa_circ_0008673 in BC plasma compared to healthy controls. Moreover, high plasma levels of this ccircRNA were positively associated with large tumor size, metastasis, and positive ER/PR status. The authors also analyzed plasma levels of hsa_circ_0008673 in paired pre- and postoperative plasma samples from 30 BC patients, and found significantly decreased expression of this ccircRNA in the postoperative samples, underlying its correlation with BC. They also demonstrated a relevant prognostic potential of hsa_circ_0008673, as its high plasma levels were associated with poor overall survival (OS) and disease-specific survival (DSS). Finally, using qRT-PCR the authors confirmed the upregulation of circ-0008673 in a panel of BC cell lines (MCF-10AT, MCF-10CA1A, MCF-10CA1H, MDA-MB-231, MDA-MB-468, MCF-7, and T47D) compared to a normal breast cell line (MCF-10A). Of note, the silencing of hsa_circ_0008673 expression in MCF-7 and MDA-MB-231 (as models of Luminal A and TNBC cancers, respectively) reduced cell proliferation and migration [67]. An in-depth mechanistic study of hsa_circ_0008673 by Lu Sun et al. demonstrated that it stimulates cell proliferation, metastasis, and angiogenesis in BC cell lines MDA-MB-231 and BT-549 by sponging miR-578 and upregulating GINS4 [68], a crucial protein for the initiation of DNA replication [69]. This study supports the finding of Youting Hu et al. [67] on the potential clinical utility of hsa_circ_0008673 for BC diagnosis and prognosis.

### 4.2. Identification of ccircRNAs by Expression Profiling of Tissue

The molecular characterization of the ccircRNA expression profile in previous studies aimed to identify new diagnostic biomarkers for BC from plasma. However, in several studies, the investigation of ccircRNAs in plasma was secondary to the goal of molecular profiling in BC tissues. For example, Xiaohan Li et al. [70] performed a microarray on three pairs of BC and adjacent tumor tissue to search for tumor-induced circRNA dysregulation. They showed 546 overexpressed and 1475 downregulated circRNAs in BC compared to normal tissue. The authors selected the six that were the most dysregulated to validate by qRT-PCR in a second sample pool of 10 pairs of BC and adjacent normal tissue. The downregulation of hsa_circ_0104824 was confirmed to be the most significant in BC. A second qRT-PCR analysis on 37 pairs of BC and adjacent normal tissue confirmed the significantly lower level of hsa_circ_0104824 in BC tissues compared to controls. Finally, they analyzed the plasma levels of hsa_circ_0104824 in 83 BC patients and 49 healthy controls by qRT-PCR, demonstrating a significantly lower expression in BC patients. Finally, the plasma concentration of hsa_circ_0104824 was found to be lower in TNBC patients than in non-TNBC patients, also suggesting that it could be used for molecular stratification of BC by liquid biopsy [70]. To date, there is no information on the functional role of hsa_circ_0104824 in BC.

Zehuan Li et al. [71] also performed a microarray assay on six paired tissues of BC and adjacent normal tissues to investigate the circRNAs and their host genes that are dysregulated in BC. After microarray analysis, the authors selected the four most upregulated circRNAs in BC and their host genes (hsa_circ_0069094 and S100P, hsa_circ_0062558 and MMP11, hsa_circ_0074026 and PITX1, hsa_circ_0079876 and ANLN), and validated them in 121 pairs of BC and adjacent normal tissues by qRT-PCR. The results confirmed the significant upregulation in BC for all of them. Similarly, the most downregulated circRNAs from microarray (hsa_circ_0017536, hsa_circ_0023302, hsa_circ_0017650, and hsa_circ_0017545) were validated by extending the analysis to 47 pairs of BC and adjacent normal tissue by qRT-PCR. The results confirmed the downregulation of these circRNAs in BC tissues compared to adjacent normal tissue. To evaluate the diagnostic potential of these circRNAs in BC, they also quantified their plasma levels in 127 BC patients and 50 healthy controls by qRT-PCR. The results showed a higher plasma expression of has_circ-0069094, has_circ-0079876, has_circ-0017650, has_circ-0017536, and of their corresponding host gene transcripts, in BC patients compared healthy controls [71]. Since has_circ-0017650 and has_circ-0017536 showed lower expression in BC tissue compared to normal tissue, and higher expression in BC plasma compared to healthy controls, further analyses are required to validate the diagnostic significance of these ccircRNAs and their biological role in the development and progression of BC. To date, of the four ccircRNAs discovered by Zehuan Li et al. [71] to be upregulated in BC plasma, only hsa_circ_0069094 has been sufficiently investigated with regard to its biological role. Several studies indicated a strong involvement of this circRNA in the regulation of glycolysis and insulin response in BC cell lines (MCF-7, MDA-MB-468, and MDA-MD-231) via the following axis: (1) miR-591/HK2 [72], an isoform of hexokinase that is part of the glycolysis pathway [73]; (2) miR-661/HMGA1 [74], a chromatin-associated protein involved in the insulin response through transcriptional regulation [75]; and (3) miR-136-5p/YWHAZ [76], a highly conserved protein that interacts with the insulin receptor substrate IRS1 and regulates the glucose receptor expression in response to insulin levels [77,78]. Moreover, Huaqing Ou et al. showed in BC cell lines (BT-549 and MDA-MB-231) that hsa_circ_0069094 stimulates cell proliferation and invasion and inhibits apoptosis by sponging miR-758-3p which, in turn, led to the overexpression of ZNF217 [79], a transcription factor regulating the expression of many tumor-related genes [80]. The large body of evidence in the literature, all related to the role of hsa_circ_0069094 as a biomarker of BC, makes this circRNA one of the most promising biomarkers for the future clinical management of BC.

The study by Yuhne Yu et al. [81] is another example of circRNA expression profiling with the aim of identifying dysregulated ccircRNAs in BC in tissue and, subsequently, validating them in plasma. The expression of the 30 most dysregulated circRNAs in BC determined by microarray assay on five BC and adjacent normal tissues, was then evaluated by qRT-PCR in 10 BC and 10 healthy control plasma samples. The expression of five ccircRNAs found to be significantly dysregulated in BC samples was further evaluated in a third cohort of 102 preoperative BC and 102 healthy control plasma samples. These two evaluation steps showed that hsa_circ_0000091 was significantly downregulated in BC plasma samples compared to healthy samples, while hsa_circ_0067772 and hsa_circ_0000512 were significantly upregulated. Interestingly, there was an increase in hsa_circ_0000091, and a decrease in hsa_circ_0067772 and hsa_circ_0000512 expression in the 102 postoperative plasma samples, compared to the corresponding preoperative plasma samples previously analyzed. All these findings support the concept that these ccircRNAs are potential predictive biomarkers for BC. A further validation step on a new cohort of 100 BC and 100 healthy plasma samples confirmed the diagnostic potential of has_circ-0000091, has_circ-0067772, and has_circ-0000512 in discriminating BC cases from healthy controls. To investigate the prognostic potential of these three ccircRNAs, the authors quantified their expression by qRT-PCR in 17 plasma samples from patients with metastatic BC at postoperative follow-up. The results showed a significant reduction in hsa_circ_0000091 plasma levels at the time of metastatic recurrence. When all 202 BC plasma samples were considered, the low expression of hsa_circ_000091 correlated positively with advanced TNM stages and axillary lymph node (ALN) metastasis, reinforcing its prognostic potential [81]. In terms of function, there is only evidence in the literature for the biological role of hsa_circ_0000512 in BC. Indeed, Li-Feng Dong et al. demonstrated in the BC cell line MDA-MB-231 that this circRNA is involved in the stimulation of PD-L1 expression, hindering T-cells cytotoxic effect via the miR-622/CMTM6 axis [82], a crucial gene for suppression of PD-L1 ubiquitination and degradation [83].

### 4.3. Identification of ccircRNAs by Literature and Database Screening

The evaluation of plasma circRNAs as BC biomarkers is not always limited to an initial circRNA expression profiling in tissue or plasma samples, but can also result from the analysis of literature data and searches in circRNA expression databases. This is the case of Jiani Liu et al. [84], who focused on hsa_circ_0000615, a circRNA derived from the pre-mRNA of ZNF609, which codes for a transcription factor protein involved in thymocyte maturation and neuron regulation [85,86]. Indeed, several studies have indicated that overexpression of hsa_circ_0000615 stimulates cell proliferation and metastasis in gastric and nasopharyngeal cancer [87,88]. To explore the involvement of this circRNA in BC, the authors tested the plasma level of hsa_circ_0000615 by qRT-PCR in 95 age-matched healthy volunteers and 95 BC patients (61 Luminal BCs, 29 HER+ BCs, and 5 TNBCs) who had undergone surgical resection of the primary tumor without neoadjuvant treatment. The results showed a significantly higher plasma level of hsa_circ_0000615 in BC patients compared to healthy controls. Moreover, the expression of this circRNA correlated positively with advanced tumor stage, metastasis, high risk of recurrence after surgery, and TNBC subtype. BC-related upregulation of hsa_circ_0000615 was confirmed in 38 paired BC and adjacent normal tissues. Finally, they showed overexpression of hsa_circ_0000615 in EVs from the culture medium of BC cell lines (MCF-7, Sk-BR-3, and 549) compared to a normal breast ductal epithelial cell line (MCF-10a), further supporting the potential of this circRNA as a biomarker for BC diagnosis [84].

Ju et al. [89] deepened the BC diagnostic potential of hsa_circ_0042881, a circRNA involved in the proliferation of glioblastoma cells [90] and the malignant promotion of gastric cancer [91]. In the first step, they quantified the expression of hsa_circ_0042881 in 56 paired BC tissues and adjacent normal tissues by qRT-PCR, and showed a significant overexpression of this circRNA in cancer tissues compared to normal tissues. Furthermore, the high levels of hsa_circ_0042881 correlated positively with the tumor size and advanced TNM stages. To validate these results, they measured plasma levels of hsa_circ_0042881 by qRT-PCR in 46 healthy volunteers and 74 BC patients without neoadjuvant treatment. As expected, the plasma levels of this ccircRNA were significantly higher in BC patients than in healthy controls. The diagnostic value of hsa_circ_0042881 for BC was also demonstrated in cell line models, where it was overexpressed in BC cell lines MCF-7 and MDA-MB-231 compared to a normal breast epithelial cell line (MCF-10A). The authors investigated the biological role of hsa_circ_0042881 in MCF-7 and MDA-MB-231 BC cells to understand its involvement in BC progression. Knocking-down and induced overexpression of hsa_circ_0042881 resulted in inhibition and stimulation of cell proliferation and malignancy, respectively, confirming the tumorigenic role of this circRNA previously observed in gastric cancer and glioblastoma. Functional studies in BC cell lines deepened the molecular pathway regulated by hsa_circ_0042881. Briefly, hsa_circ_0042881 sequesters mir-217 and, subsequently, leads to overexpression of SOS1, activating RAS and the associated MAPK and PIK3CA/AKT pathway which, in turn, leads to an increase in cell proliferation and invasion [89]. Further investigation of the hsa_circ_0042881-regulated pathway in BC tissues could also be of clinical interest for targeted therapies.

Bo Fu et al. [92] investigated the biological role of hsa_circ_0001944 in promoting BC invasion and breast cancer brain metastases (BCBM), and its potential as a prognostic biomarker for BC. The choice of this circRNA stems from a previous study by the same authors, in which hsa_circ_0001944 was found to be one of the most upregulated circRNAs in the brain metastatic 231-BR cells compared to the non-organ-specific metastatic MDA-MB-231 parental cells from which they are derived [92]. By qRT-PCR, they quantified the expression levels of hsa_circ_0001944 in 13 paired non-metastatic BC and adjacent normal tissues and in 6 BCBM tissues. The results showed a significantly higher expression of hsa_circ_0001944 in BCBM tissues compared to non-metastatic BC or normal tissues. They also analyzed the plasma levels of hsa_circ_0001944 in 20 BCBM and 20 non-metastatic BC patients. Again, the ccircRNA was found to be upregulated in BCBM patients compared to non-metastatic BC patients. Finally, they tested the prognostic potential of hsa_circ_0001944 in the primary tumor tissues of 53 BCBM patients. They found that patients with higher levels of hsa_circ_0001944 had shorter brain metastasis-free survival (BMFS) than those with lower hsa_circ_0001944 expression, confirming the prognostic potential of this circRNA. Of note, Bo Fu et al. [93] did not perform a similar plasma analysis to investigate whether this circRNA could be a biomarker in liquid biopsy. In terms of function, they have demonstrated that hsa_circ_0001944 acts as a miRNA sponge to inhibit miR-125a, leading to activation of the SHH pathway with downstream stimulation of BRD4 and MMP9 [93], crucial genes for tumor invasion and metastasis [94,95].

### 4.4. TNBC-Associated ccircRNAs

The analysis of plasma circRNAs was also helpful for the stratification of BC, as already mentioned above. TNBC is a rare subtype of BC characterized by a high risk of recurrence and metastasis [96]. For this reason, it represents a challenge for new biomarkers and clinical treatments. Chen et al. [97] focused on the study of hsa_circ_0004623, which is linked to the HIF1A gene encoding an important transcriptional regulator of the hypoxia response, and its clinical relevance in the management of TNBC [98,99]. The authors first analyzed the levels of hsa_circ_0004623 in 50 paired BC tissues and adjacent normal tissues by qRT-PCR. The results showed a higher expression of this circRNA in BC tissues than in normal tissues; moreover, its expression was negatively correlated with hormone receptor status (HER2, PR, and ER), suggesting that this circRNA is a TNBC-associated biomarker. Moreover, high levels of hsa_circ_0004623 were strongly associated with metastasis and poor OS. Given the relevance of hsa_circ_0004623 in TNBC, they tested hsa_circ_0004623 as a new potential diagnostic biomarker. To this end, they compared the levels of EVs hsa_circ_0004623 from MDA-MB-231 cells treated with a plasmid for the constitutive expression of hsa_circ_0004623 and MDA-MD-231 control cells treated with the empty plasmid vector pLCDH. The qRT-PCR results indicated higher levels of hsa_circ_0004623 in EVs from MDA-MD-231 cells transformed by plasmid containing circRNA than in those transformed with the empty vector. Interestingly, the non-transformed MDA-MB-231 cells treated with EVs containing hsa_circ_0004623 increased their proliferation rate compared to the MDA-MB-231 cells untreated with EVs. They quantified the plasma levels of hsa_circ_0004623 in 24 BC patients and 68 age-matched healthy controls by qRT-PCR. In agreement with the finding in tissues, BC patients showed a higher plasma expression of this ccircRNA than the healthy controls. Finally, they performed several functional analyses on MDA-MB-231 and MDA-MB-468 cell lines to elucidate the biological role of hsa_circ_0004623 in BC, and showed that it acts as a proliferation and invasion enhancer. This circRNA was found to sponge miR-149-5p, leading to expression of NFIB, activation of the AKT/STAT pathway, and inhibition of p21. Moreover, NFIB activates transcription of the RNA-binding protein FUS (a transcriptional regulator also involved in the DNA repair mechanism [100]) which, in turn, stimulates the biogenesis of hsa_circ_0004623, thus determining a positive feedback loop [97].

Similarly, to Chen et al. [97], Darbeheshti et al. [101] focused their study on dysregulated circRNAs in TNBC to discover novel biomarkers. They analyzed has_circ_0000977, one of the most upregulated circRNAs in BC, according to the circRNA expression dataset GSE101124. They analyzed the expression of has_circ_0000977 on tissues from 40 TNBC, 20 Luminal A, 18 Luminal B, and 17 HER2+ BC patients by qRT-PCR. The results showed that the levels of has_circ_0000977 were significantly lower in TNBC patients than in non-TNBC patients. In addition, low expression of has_circ_0000977 was associated with young patients, short disease-free survival (DFS), and large tumor size. This downregulation was also confirmed in cell lines, where higher levels of has_circ_0000977 were found in non-TNBC cell lines (MCF7, MADA-MB-361, and SKBR3) than in TNBC cell lines (MDA-MB-231 and MDA-MB-468). To further investigate the clinical potential of this circRNA, the authors quantified its plasma levels in 20 TNBC patients and 20 age-matched healthy volunteers by qRT-PCR. In agreement with a previous finding, plasma levels of hsa_circ_0000977 were lower in TNBC patients compared to healthy controls. Functionally, has_circ_0000977 appears to act as a miRNA sponge for miR-135b-5p which, in turn, can regulate APC, GATA3, and the β-catenin pathway to increase cell proliferation in the MDA-MD-231 TNBC cell line [101].

Song et al. [102] focused on the characterization of autophagy-associated circRNAs in TNBC, as TNBC cells have been reported to have higher autophagy activity compared to non-TNBC cells [103]. First, the authors performed RNA-seq on three groups of MDA-MB-231 cells: untreated, treated with an amino acid-free culture medium (to stimulate autophagy), and treated with chloroquine (an autophagy inhibitor). The results showed that 22 circRNAs were dysregulated in cells treated with an amino acid-free culture medium, compared to untreated and chloroquine-treated cells. Among these, they selected hsa_circ_0080222 as the most significantly dysregulated circRNA in cells with increased autophagy activity. The authors analyzed the expression levels of hsa_circ_0080222 in different cell lines by qRT-PCR, showing a higher expression in TNBC cell lines (HCC1937, MDA-MB-231, MDA-MB-488, and CAL-51) compared to non TNBC cell lines (T47D, BT474, MCF7, and ZR-75-1). They validated the expression levels of hsa_circ_0080222 in 85 paired BC tissues (9 HER2+, 45 Luminal, 31 TNBC) and adjacent normal tissues by qRT-PCR. In addition, hsa_circ_0080222 was found to be upregulated in tumor tissues compared to normal tissues, but this difference was only significant in the TNBC subgroup. Moreover, BC tissues from patients with lymph node metastasis showed a higher level of hsa_circ_0080222 than those from patients without metastasis. This suggests a role for this circRNA in tumor progression, in addition to the autophagy induction and its potential as a TNBC prognostic biomarker. Furthermore, they quantified hsa_circ_0080222 levels in EVs from 74 BC plasma and 24 healthy control plasma, showing that hsa_circ_0080222 levels were higher in EVs from BC compared to healthy controls. Functionally, hsa_circ_0080222 acts as a miRNA sponge for miR-244-5p in the MDA-MB-231 cell line, and reduces its inhibitory effect on the expression of ATG13 and ULK1, whose protein products are crucial for phagosome formation [104,105]. Finally, hsa_circ_0080222 was found to bind ANXA2 to release TFEB, a transcription factor that induces the expression of autophagy-related genes such as ATG9B, LC3, and p62 [102,106].

Jiulong Ma et al. [107] further explored the clinical value of hsa_circ_0080220 in TNBC and its response to treatment. The choice of this circRNA, derived from a previous study, performed by the same research group, indicating a possible role of hsa_circ_0080220 in TNBC response to pirarubicin (THP) [107], a chemotherapeutic agent for the treatment of BC [108]. They measured the expression of hsa_circ_0080220 in 38 paired TNBC tissues and adjacent normal tissues using qRT-PCR. The results showed a significant upregulation of this circRNA in TNBC tissues compared to normal tissues. Moreover, the high levels of hsa_circ_0080220 were associated with poor OS and disease-free survival (DFS) in the 38 TNBC patients analyzed. Further qRT-PCR analysis on BC cell lines showed that hsa_circ_0080220 was overexpressed in TNBC cell lines (MDA-MB-231, MDA-MB-468, and HCC-1806) compared to non-TNBC cell lines (MCF-7, BT-474, and SKBR-3) and to a normal breast epithelial cell line (MCF-10A). Moreover, the expression of hsa_circ_0080220 in the TNBC MDA-MB-231 cell line correlated positively with THP resistance, invasion, and migration. The authors also measured the expression of hsa_circ_0080220 on plasma EVs from 40 non-TNBC, 49 TNBC, and 10 neo-adjuvant (NAC)-treated TNBC. The results showed a significant overexpression of this ccircRNA in TNBC patients compared to non-TNBC patients, and a significant downregulation in NAC-treated TNBC patients compared to untreated TNBC patients. In functional terms, Jiulong Ma et al. [107] confirmed the involvement of hsa_circ_0080220 in the upregulation of its host gene EGFR through the sponging of miR-1299 which, in turn, leads to the stimulation of cell proliferation and invasion via the MAPK/AKT signaling pathway [109].

Similarly, Fan Zhang et al. [110] focused on the identification of circRNAs related to TNBC and to the response to THP. Through bioinformatic analyses, Fan Zhang et al. [110] identified hsa_circ_0000854 as a potential candidate for their investigation and validation on cell line models and clinical samples. First, they quantified the expression of hsa_circ_0000854 on THP-treated MDA-MB-231 cells and untreated cells by qRT-PCR. The results showed a significant downregulation of this circRNA in treated cells compared to untreated cells. In addition, a further qRT-PCR analysis showed overexpression of hsa_circ_0000854 in TNBC cell lines (MDA-MB-231, MDA-MB-468, and MDA-MB-436) compared to non-TNBC cell lines (MCF-7 and SK-BR-3) and a normal breast epithelial cell line (MCF-10A). Knockdown of hsa_circ_0000854 in TNBC MDA-MB-231 resulted in increased resistance to the cytotoxic activity of THP. The authors quantified hsa_circ_0000854 levels in 34 paired TNBC tissues and adjacent normal tissues and in 24 non-triple negative BC tissues, and found overexpression of this circRNA in TNBC tissues compared to non-triple negative BC tissues and healthy tissues. Moreover, TNBC patients with higher hsa_circ_0000854 expression had a higher risk of developing metastasis. Finally, the TNBC-related overexpression in plasma EVs was confirmed from 42 TNBC and 35 non-TNBC patients. From a functional perspective, overexpression of hsa_circ_0000854 triggers activation of the MAPK signaling pathway through sponging of miR-1200 and subsequent overexpression of TPR, a protein involved in nuclear export of mRNAs and proteins [110].

Table 1 summarizes the BC-related ccircRNAs from plasma whose function is known. Table 2 summarizes the BC-related ccircRNAs with unknown functions. As the quantification of circRNAs in plasma or tissues was performed in two different cohorts in the reviewed articles, we have only reported the clinical value of circRNAs tested on the plasma cohort in Table 1 and Table 2.

## 5. Conclusions

Nowadays, circRNAs appear to be more and more important components of a complex gene regulatory network that may play a key role in the development and progression of many diseases, such as cancer. The fact that they can regulate miRNA activities by preventing their binding to targets emphasizes the importance of circRNA in cell biology and pathology.

Of the 14 articles analyzed, 12 critically evaluated circRNA expression in both plasma and tissues or cell lines. The comparison showed a general agreement between liquid biopsy and tissue/cell samples, emphasizing the relevance of circulating circRNA (ccircRNAs) for translational research. The only exceptions were hsa_circ_0017650 and hsa_circ_0017536, where there was no match in the tissues. In this case, these circRNAs are appear to be only an expression of circulating molecules related to tumor cell death or tumor cell secretion activities. There are several explanations for this phenomenon, e.g., the type of circRNA secretion in the blood. Indeed, plasma circRNAs can arise by passive cell death induced by drugs, hypoxia, inflammation, or endogenous molecular pathways, or they can also be actively secreted into the bloodstream packaged in EVs or TEPs [111]. This implies that the comparison of the circRNA profile of plasma and tissue depends on active/passive molecular secretion and tumor/tissue integrity. Moreover, Zehuan Li et al. showed concordance of hsa_circ_0069094 and hsa_circ_0079876 presence between plasma and tissue, but not for hsa_circ_0017650 and hsa_circ_0017536 [71], suggesting a mode of secretion specific to each circRNA in the bloodstream. Since hsa_circ_0017650 and hsa_circ_0017536 are less expressed in tumor tissues, we could speculate about a possible oncosuppressive role of these circRNAs in BC. It could be that the identification of ccircRNAs with a tumor-suppressive role in plasma is more difficult because their expression is repressed in cancer cells. However, this is an empirical idea that requires further investigations to gain deeper insights.

A total of 1569 BC (with a minimum of 20 and a maximum of 424) and 740 healthy plasma samples (with a minimum of 20 and a maximum of 212) were analyzed. In all articles, the plasma expression levels of each circRNA were determined by qRT-PCR, and the circRNA to be tested was selected by microarray, NGS, bioinformatics, or literature search. The analyzed articles show several examples of ccircRNAs with potential diagnostic, predictive, and prognostic value. Moreover, several diagnostic ccircRNA biomarkers were correlated with hormone receptor status, and could be useful for better stratification of BC diagnosis. The reported predictive ccircRNAs are all associated with the evaluation of their postoperative and post-NAC levels, and no article reported ccircRNAs associated with treatment resistance in BC. Fan Zhang et al. and Jouling Ma et al. have shown how has_circ_0080220 and has_circ_0000854 are involved in THP resistance in TNBC cell lines, but a more in-depth study on plasma is needed [109,110]. The investigation of ccircRNAs as predictive biomarkers of BC response to the most common therapy could be an interesting future perspective.

The evaluation of the biological role of circRNAs is complex due to their dynamic formation and functional versatility. For example, hsa_circ_0006404 (which Lin et al. found to be overexpressed in BC plasma compared to healthy plasma [66]) stimulates the progression of hepatocellular carcinoma (HCC) by sponging miR-624 [112], the progression of prostate cancer (PC) via the miR-1299/CFL2 [113] and miR-29a-3p/SLC25A15 axis [114], increasing the resistance of ovarian cancer (OC) to docetaxel treatment via the miR-346/DKK3/p-GP axis [115], and inhibiting the progression of colorectal cancer (CC) via the miR-543/LATS1 axis [116]. This suggests that a circRNA may have tumor-specific activity and intra-tumor heterogeneity in regulating different signaling pathways. In addition, Song et al. and Jouling Ma et al. described two circRNAs derived from the same host gene EGFR: hsa_circ_0080222 (consisting of 4–11 exons of EGFR) and hsa_circ_0080220 (consisting of 2–4 exons of EGFR) [102,109]. These circRNAs regulated different signaling pathways (autophagy for hsa_circ_0080222 and MAPK/AKT for hsa_circ_0080220) and had different mechanisms of action (miRNA sponge and cRBP for hsa_circ_0080222 and miRNA sponge for hsa_circ_0080220). The complexity of the circRNA regulatory network emphasizes the utility of appropriate functional investigation. Overall, most reported ccircRNAs are involved in the regulation of BC cell proliferation and invasion, particularly via the MAPK/AKT signaling pathway, which is frequently deregulated in BC. Metabolism (hsa_circ_0069094 and hsa_circ_0080222), cellular signaling (hsa_circ_0108942) and tumor response to the immune system (hsa_circ_0000512) may be other pathways regulated by ccircRNAs that are of clinical interest in BC.

Figure 5 and Figure 6 shows an overview of the dysregulated circRNAs that have been found in plasma and whose functions are known. It is evident that most of the upregulated circRNAs promote cell proliferation, survival, and invasion, while the downregulated ones promote apoptosis and cytoskeletal modulation. Interestingly, two circRNAs are involved in metabolic pathways related to cancer death (autophagy) or survival (glycolysis). Finally, two circRNAs have been found to be associated with the mechanism by which the tumor evades recognition by the immune system.

Unfortunately, the ccircRNAs mentioned in the articles discussed have not been shown to be potential biomarkers in more than one study, and the function of many of them remains to be further demonstrated. However, based on the findings from the literature, some ccircRNAs appear to be promising for future clinical use.

According to ClinicalTrial.gov data, there is one observational study (ClinicalTrials.gov ID: NCT05771337) that aims to evaluate the diagnostic and prognostic value of hsa_circ_0001785 (Circ-ELP3) and hsa_circ_100219 (Circ-FAF1) in serum samples from BC patients. The study also focuses on the evaluation of serum chemerin, a leukocyte chemoattractant, as a diagnostic biomarker that can be coupled with hsa_circ_0001785 and hsa_circ_100219 in plasma [117,118]. This clinical trial is in the recruitment phase and the authors estimate that the trial will be completed in 2025.

For example, Lin et al. pointed out hsa_circ_0006404 as an evaluable BC plasma biomarker [66], and several studies showed that it is involved in the progression of HCC, PC, OC, and CC [112,113,114,115,116], although nothing has been reported about its role in BC. Zehuan Li et al. showed significant upregulation of hsa_circ_0069094 in BC plasma compared to healthy controls, and several independent studies confirm its oncogenic role in BC [71]. However, circRNAs and their biological function are a relatively new field of research, where much remains to be done. According to recent findings, circRNAs are characterized by a highly complex functional network associated with a dynamic and multi-layered biosynthetic process. In order to properly assess the clinical utility of circRNAs, these signaling pathways need to be studied in detail, as mentioned above.

Although knowledge about the role of circRNAs in BC is still in its infancy, there are many indications that they could be useful as biomarkers for the diagnosis and prognosis of this disease. The identification of circRNAs from an RNA-seq dataset using various algorithms and bioinformatics tools is an important and challenging step in the study of circRNA. Despite the development of several pipelines for the bioinformatic detection of circRNAs, some problems still need to be solved. For example, the alignment of short-read fragments could be affected by ambiguous read mapping events, especially for genomic sites with similar sequences. In addition, the lack of a gold standard for RNA library preparation, RNA-seq approach and bioinformatics pipeline hinders the appropriate and reproducible identification of circRNAs on a genome-wide scale [119].

In the coming years, in-depth analyses of the biological role at the cellular level and secretion modes of ccircRNAs in the bloodstream by specifically applied high-throughput technologies and optimized bioinformatic analyses could open up the understanding of their involvement in BC and their clinical utility as biomarkers or targets for therapies.

## Figures and Tables

**Figure 1 biomedicines-12-00875-f001:**
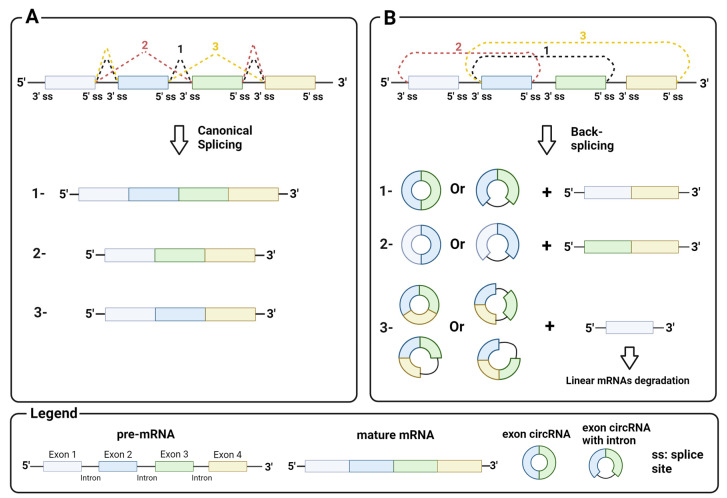
Mechanism of back-splicing: (**A**) canonical splicing with a 5′ upstream donor splice site and a 3′ downstream acceptor splice site and an example of products (1, mature mRNA with all exons; 2 and 3, alternative mature mRNA); (**B**) back-splicing with a 3′ downstream donor splice site and a 5′ upstream acceptor splice site and an example of products. Image created with Biorender.

**Figure 2 biomedicines-12-00875-f002:**
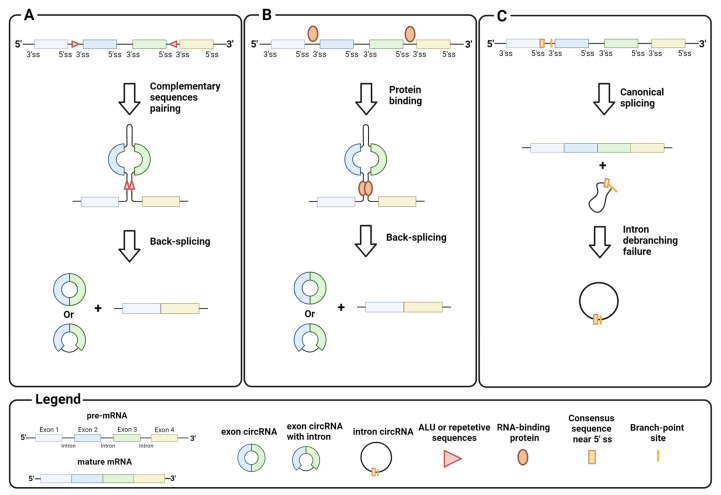
Biogenesis of circRNA: (**A**) back-splicing of introns, triggered by the pairing of complementary sequences, generates circRNAs that contain exons; (**B**) back-splicing of introns, triggered by interactions between intron-binding proteins, generates circRNAs that contain exons; (**C**) intron debranching defects prevent lariat degradation that becomes an intron circRNA. Image created with Biorender.

**Figure 3 biomedicines-12-00875-f003:**
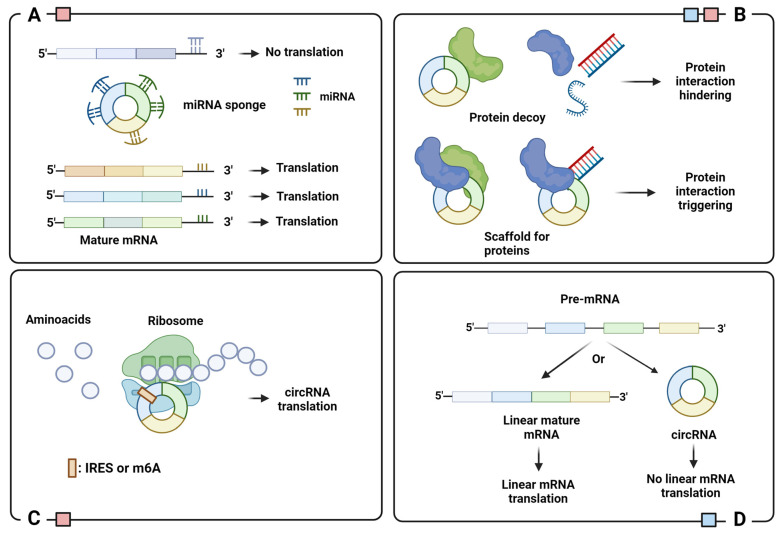
Schematic representation of circRNA functions: (**A**) miRNA sponge; (**B**) protein binding regulator; (**C**) template for translation; (**D**) translation regulator. Legend: IRES, Internal Ribosome Entry Site; m6A, N6-Methyladenosine. Colored squares indicate the cellular localization of circRNAs with indicated function: blue for nucleus and red for cytoplasm. Image created with Biorender.

**Figure 4 biomedicines-12-00875-f004:**
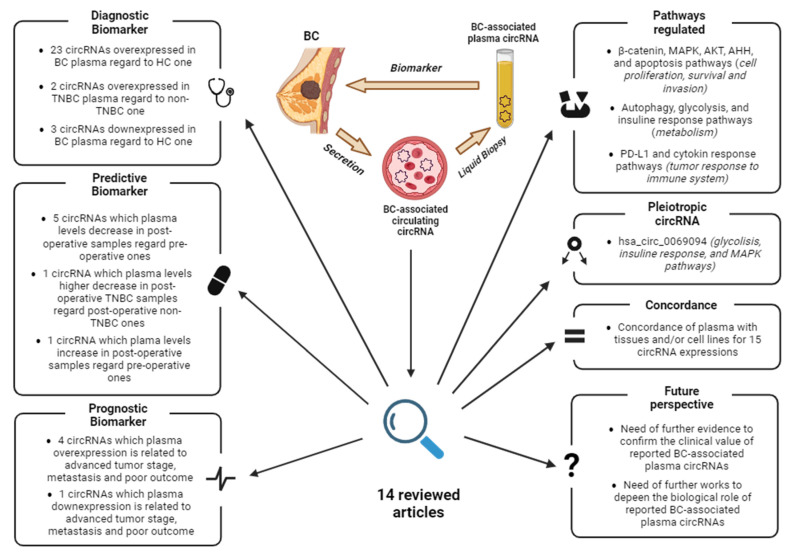
Graphical abstract summarizing the key points and findings of this review. Abbreviations: BC, Breast Cancer; HC, healthy controls; TNBC, triple-negative breast cancer. Image created with Biorender.

**Figure 5 biomedicines-12-00875-f005:**
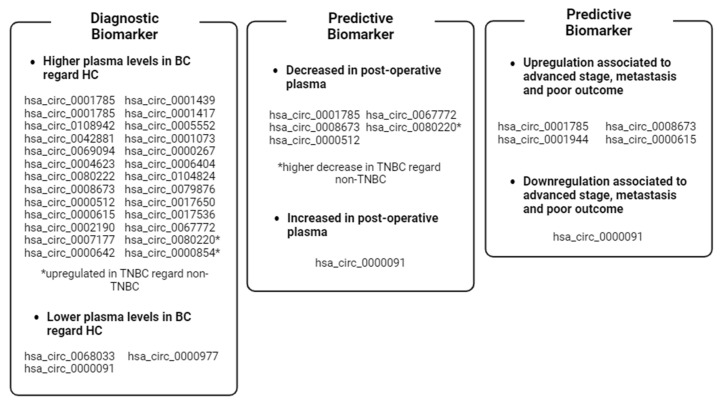
Recap of the clinical value of BC-associated plasma ccircRNAs reported in the review. * circRNAs dysregulated in TNBC cases. Image created with Biorender.

**Figure 6 biomedicines-12-00875-f006:**
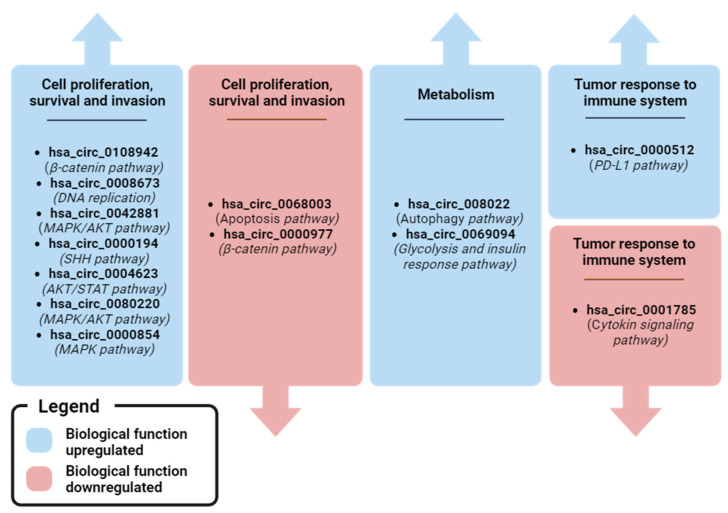
Recap of biological pathways regulated by plasma ccircRNA with clinical relevance in BC. Image created with Biorender.

**Table 1 biomedicines-12-00875-t001:** Recap of plasma ccircRNAs with clinical value in BC and known biological role. Abbreviations: AUC, area under curve; BC, breast cancer; BMBC, brain metastasis-breast cancer; DSS, disease-specific survival; NAC, neo-adjuvant chemotherapy; NR, not reported; OS, overall survival.

Cit.	Sample (*n*)	circBase ID (Host Gene)	Clinical Value	AUC	Findings	Expression Match with Tissues and Cell Lines	Biological Function (Pathway Regulated)
		hsa_circ_0001785 (ELP3)	Diagnostic, Prognostic, Predictive	0.771	Higher levels in BC patients than in healthy controls. High levels were associated with advanced tumor stages and metastasis. Levels decrease in post-operative samples compared to matched pre-operative ones	NR	miRNA sponging: miR-942/SOCS3 axis (cytokine signaling pathway)
Yin et al. [60]	Pre-operative BC (83): 57 matched post-operative; Healthy controls (25)	hsa_circ_0108942 (ANKRD12)	Diagnostic	0.701	Higher levels in BC patients than in healthy controls	NR	miRNA sponging: miR-1178-3p/TMED3 axis (β-catenin pathway)
		hsa_circ_0068033 (NAALADL2)	Diagnostic	0.619	Lower levels in BC patients than in healthy controls.	NR	miRNA sponging: miR-659 (Apoptosis pathway)
Ju et al. [89]	Pre-operative BC (74); Healthy controls (46)	hsa_circ_0042881 (NF1)	Diagnostic	0.802	Higher levels in BC patients than in healthy controls	Yes, with tissues	miRNA sponging: miR-217/SOS1 axis (MAPK/AKT pathway)
Zehuan Li et al. [71]	Pre-operative BC (127); Healthy controls (50)	hsa_circ_0069094 (S100P)	Diagnostic	0.680	Higher levels in BC patients than in healthy controls	Yes, with tissues and cell lines	miRNA sponging: miR-591/HK2 axis (Glycolysis), miR-136-5p/YWHAZ axis (insulin response pathway), miR-758-3p/ZNF217 axis (MAPK/AKT pathway), miR-661/HMGA1 axis (glycolysis, insulin response pathways)
Chen et al. [97]	BC (24); Healthy controls (68)	hsa_circ_0004623 (HIF1A)	Diagnostic	0.897	Higher levels in BC patients than in healthy controls	Yes, with tissues and cell lines	miRNA sponging: miR-149-5p/NFIB axis (AKT/STAT pathway)
Darbeheshti et al. [101]	TNBC (20); Healthy controls (20)	hsa_circ_0000977 (NOL10)	Diagnostic	0.960	Lower levels in TNBC patients than in healthy controls	Yes, with tissues and cell lines	miRNA sponging: miR-135b-5p/APC/GATA3 axis (B-catenin pathway)
Song et al. [102]	BC (74); Healthy controls (24)	hsa_circ_0080222 (EGFR)	Diagnostic	NR	Higher levels in BC patients than in healthy controls	Yes, with tissues and cell lines	miRNA sponging: miR-224-5p/ATG13/ULK1 axis, protein sponging: ANXA2/TFEB axis (Autophagy pathway)
Jiulong et al. [109]	TNBC (49); non-TNBC (40); TNBC with NAC (10)	hsa_circ_0080220 (EGFR)	Diagnostic, Predictive	NR	Higher levels in TNBC patients than in non-TNBC ones. Lower levels in NAC-treated TNBC patients than in untreated TNBC ones	Yes, with tissues and cell lines	miRNA sponging: miR-1299/EGFR axis (MAPK/AKT pathway)
Fan Zhang et al. [110]	TNBC (42); non-TNBC (35)	hsa_circ_0000854 (ZCCHC2)	Diagnostic	NR	Higher levels in TNBC patients than in non-TNBC ones	Yes, with tissues and cell lines	miRNA sponging: miR-1200/TPR axis (MAPK pathway).
Bo Fu et al. [93]	BMBC (20); non-metastatic BC (20)	hsa_circ_0001944 (FIRRE)	Prognostic	NR	Higher levels in BMBC patients than in non-metastatic BC ones	Yes, with tissues and cell lines	miRNA sponging: miR-125a/BRD4 axis (SHH pathway)
Youting et al. [67]	Pre-operative BC plasma (384): 30 matched post-operative; Healthy controls (108)	hsa_circ_0008673 (BRCA1)	Diagnostic, Prognostic, Predictive	0.833	Higher levels in BC patients than in healthy controls. High levels are associated with large tumor size, metastasis, poor OS and DSS, and hormone receptor status. Levels decrease in post-operative samples compared to pre-operative ones	Yes, with cell lines	miRNA sponging: miR-578/GINS4 axis (DNA replication pathway)
Yuhne et al. [81]	Pre-operative BC (212): 102 matched post-operative; Healthy controls (212)	hsa_Circ_0000512 (RPPH1)	Diagnostic, Predictive	0.704	Higher levels in BC patients than in healthy controls. Levels decrease in post-operative samples compared to pre-operative ones.	Yes, with tissues	miRNA sponging: miR-622 (PD-L1 pathway)

**Table 2 biomedicines-12-00875-t002:** Recap of plasma ccircRNAs with clinical value in BC and unknown biological role. Abbreviations: AUC, area under curve; ALN, axillary lymph node; BBT, benign brain tumor; BC, breast cancer; NR, not reported; TNBC (triple-negative breast cancer).

Cit.	Sample (*n*)	circBase ID (Host Gene)	Clinical Value	AUC	Findings	Expression Match with Tissues and Cell Lines
Jiani Liu et al. [84]	BC (95); Healthy controls (95)	hsa_circ_0000615 (ZNF609)	Diagnostic, Prognostic, Predictive	0.904	Higher levels in BC patients than in healthy controls. High levels are associated with advanced tumor stage, metastasis, high recurrence risk after surgery and TNBC subtype	Yes, with tissues and cell lines
		hsa_circ_0002190 (KLHDC10)				
		hsa_Circ_0007177 (CCZ1)				
		hsa_circ_0000642 (ZFAND6)				
		hsa_circ_0001439 (SCLT1)				
Lin et al. [66]	BC (158); Healthy controls (43); BBT (88)	hsa_circ_0001417 (ANKRD17)	Diagnostic	0.83 (for the panel)	Higher levels in BC patients than in healthy controls and BBT	NR
		hsa_circ_0005552 (EHBP1)				
		hsa_circ_0001073 (ACVR2A)				
		hsa_circ_0000267 (FAM53B)				
		hsa_circ_0006404 (FOXO3)				
Xiaohan Li et al. [70]	BC (83); Healthy controls (49)	hsa_circ_0104824 (NTRK3)	Diagnostic	0.849	Higher levels in BC patients than healthy controls	Yes, with tissues
		hsa_circ_0079876 (ANLN)		0.622		Yes, with tissues
Zehuan Li et al. [71]	BC (127); Healthy controls (50)	hsa_circ_0017650 (ITIH5)	Diagnostic	0.758	Higher levels in BC patients than in healthy controls	No, with tissues
		hsa_circ_0017536 (AKR1C1)		0.615		No, with tissues
Yuhne et al. [81]	Pre-operative BC (212): 102 matched postoperative; Healthy controls (212)	hsa_circ_0000091 (RPAP2)	Diagnostic, Prognostic, Predictive	0.825	Lower levels in BC patients than in healthy controls. Levels increase in post-operative samples compared to pre-operative ones. Low levels are associated with advanced TNM stages and ALN metastasis	Yes, with tissue
		hsa_Circ_0067772 (SLC33A1)	Diagnostic, Predictive	0.730	Higher levels in BC patients than in healthy controls. Levels decrease in post-operative samples compared to pre-operative ones	Yes, with tissues

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
