# Peer review of "Plasma Circular RNAs as Biomarkers for Breast Cancer"

_biomedicines, 2024, doi:10.3390/biomedicines12040875_

Round 1

Reviewer 1 Report

Comments and Suggestions for Authors

This review describes the latest findings on BC-associated ccircRNAs in plasma and their clinical utility, emphasizing the potential of ccircRNAs as BC biomarkers, particularly in diagnosis. The study of circulating circRNAs (ccircRNAs) in plasma could provide new non-invasive diagnostic, prognostic, and predictive biomarkers for BC. The reported BC-associated ccircRNAs are mainly involved in regulating cell survival, proliferation, and invasion via the MAPK/AKT signaling pathways. This study is significant for breast cancer precision treatment. However, the following issues are required for explaining:

1. The review should further explore the potential of Plasma Circular RNA as a predictive marker for breast cancer drug sensitivity. The authors should discuss ongoing clinical trials investigating Plasma Circular RNA as a screening tool for breast cancer drug sensitivity would provide a comprehensive overview of current research in this area. An additional table is recommended.

2. The authors should add a summary table presenting the main ccircRNAs discussed in the review, along with their reported functions and potential as biomarkers for breast cancer.

3. The authors should add a Figure (or Graphical abstract) to summarize the main findings and key points of the review.

4. Table 1: The specificity and sensitivity of Plasma Circular RNA should be summarized.

5. Some similar studies regarding the Circular RNA should be cited and discussed. For example, PMID: 36672544, 38130633, 37040525.

6. The authors are recommended to consider engaging a professional language editing service to ensure the clarity and coherence of the manuscript.

Comments on the Quality of English Language

Moderate editing of English language required

Author Response

We thank the Reviewer for useful suggestions and comments. All changes to the text are highlighted in azure blue (necessary additions) and yellow (English editing). Please note that the lines indicated refer to the clean PDF version of the manuscript.

This review describes the latest findings on BC-associated ccircRNAs in plasma and their clinical utility, emphasizing the potential of ccircRNAs as BC biomarkers, particularly in diagnosis. The study of circulating circRNAs (ccircRNAs) in plasma could provide new non-invasive diagnostic, prognostic, and predictive biomarkers for BC. The reported BC-associated ccircRNAs are mainly involved in regulating cell survival, proliferation, and invasion via the MAPK/AKT signaling pathways. This study is significant for breast cancer precision treatment. However, the following issues are required for explaining:

  1. The review should further explore the potential of Plasma Circular RNA as a predictive marker for breast cancer drug sensitivity. The authors should discuss ongoing clinical trials investigating Plasma Circular RNA as a screening tool for breast cancer drug sensitivity would provide a comprehensive overview of current research in this area. An additional table is recommended.

As requested, we have added the information on the only ongoing clinical trial of BC-associated circRNAs provided by ClinicalTrials.gov in the "Conclusions" section (lines 677-683):

 “According to ClinicalTrial.gov data, there is one observational study (ClinicalTri-als.gov ID: NCT05771337) that aims to evaluate the diagnostic and prognostic value of hsa_circ_0001785 (Circ-ELP3) and hsa_circ_100219 (Circ-FAF1) in serum samples from BC patients. The study also focuses on the evaluation of serum chemerin, a leukocyte chemoattractant, as a diagnostic biomarker that can be coupled with hsa_circ_0001785 and hsa_circ_100219 in plasma [117,118]. This clinical trial is in the recruitment phase and the authors estimate that the trial will be completed in 2025.”

  1. The authors should add a summary table presenting the main ccircRNAs discussed in the review, along with their reported functions and potential as biomarkers for breast cancer.

We have added a new Figure 5 to the “Conclusions” section that summarises the clinical value of the BC-associated ccircRNAs discussed in the review (line 708). Instead, the functions of the circRNAs discussed have been presented in Figure 6 (line 708).

  1. The authors should add a Figure (or Graphical abstract) to summarize the main findings and key points of the review.

In the section “Liquid biopsy and circulating circRNAs”, we have added a Graphical abstract resuming the most important points and results of the review (line 244).

  1. Table 1: The specificity and sensitivity of Plasma Circular RNA should be summarized.

 We have added a new column in Tables 1 and 2 showing the AUC value of the circRNAs in plasma discussed in the review. If the AUC value was not given in the original article, we have added the abbreviation NR (Not Reported).

  1. Some similar studies regarding the Circular RNA should be cited and discussed. For example, PMID: 36672544, 38130633, 37040525.

As requested, we have cited and discussed the proposed articles in the “Biological functions” section (lines 178-190):

 Increasing evidence for the importance of circRNAs in human disease emphasises their potential as efficient biomarkers and novel therapeutic targets. One example is the study by Galardi et al. describing the crucial role of circRNAs in paediatric tumours [42]. Among the most promising circRNAs reported, circSKA3 plays a crucial role in promoting cell proliferation in medulloblastoma, one of the most common brain tumours in children, by sponging miR-383-5p, miR-326 and miR-520-h [43–45]. Similarly, hsa_circ_0000527 and hsa_circ_0000034 stimulate the proliferation and invasion of retinoblastoma tumour cells by sponging miR-646 and miR-361-3p, respectively [46–48]. Exon skipping is an interesting new therapeutic approach based on the removal of aberrant exons and the production of truncated proteins to partially restore the physiological cellular phenotype. [49]. In addition, the potential clinical applications of circRNAs have been extended to their use as novel nucleic acid vaccines, but many manufacturing and safety limitations still need to be overcome [50].”

  1. The authors are recommended to consider engaging a professional language editing service to ensure the clarity and coherence of the manuscript.

As recommended, we have reviewed and revised the manuscript to correct the English language and ensure clarity and coherence. The changes are highlighted in yellow.

Reviewer 2 Report

Comments and Suggestions for Authors

Well-written review on Plasma cirRNA as Biomarkers for Breast Cancer .. . There is an unmet need to develop early biomarkers for all cancers in general, BC in particular, due to the severe nature of this disease. The review has done some good graphic representation, and I suppose everything was done under the Biorender /similar tools with proper agreement and license. I have several concerns about the review in its present form. These are summarized below:

1.       The title itself: The portion “ State of art and clinical perspective,” I found hardly there is any discussion of the state-of-the-art tools or technology described in this review. This extra add-on seems unnecessary and largely irrelevant. It may be deleted from the title

2.       The very first section on Page # 1 failed to justify why we need to study cirRNA for BC specifically. Why is the present targets in BC, as well as the existing tools/methodology/technology, lacking, and hence we need to focus on rRNA? BC is perhaps one of the most well-explored cancers, with thousands of sequencing data on mRNA, exome, mutations, etc, well documented. How our focus on cirRNA may add value to this existing knowledge.

3.       Section 3, Page 6  ( Liquid Biopsy) section needs to be reframed and justified. The first line “ Liquid biopsy refers to a class of biological fluids of great clinical relevance..” . seems incorrect as liquid biopsy is not a class of biological fluids.. it’s a technique.

4.       Why do the authors just want to focus on liquid biopsy as the only means for detecting cirRNA? cirRNA can also be assayed in cellular material? Please elaborate on biological samples where cirRNA can be assayed

5.       I was expecting a section where the authors discuss the bioinformatics aspect of detecting cirRNA  as this is a challenging task and worth discussing.

6.       section 4 : BC-associated ccircRNA is very long and needs to be fragmented into 4-5  subsections like : maybe cell line-based reporting vs primary cancer-based reporting or cancer stage-specific ccirRNA expression, the role of cirRNA in cancer immune evasion, drug resistance. Rather than one long description, it will be helpful to break in subtopics as discussed

7.       A section on diagnostic challenges and their remedy in detecting cirRNA needs to be added

8.       A graphical summary explaining the pleiotropic role of cirRNA in BC is recommended

Comments on the Quality of English Language

N/A

Author Response

 We thank the Reviewer for useful suggestions and comments. All changes to the text are highlighted in azure blue (necessary additions) and yellow (English editing). Please note that the lines indicated refer to the clean PDF version of the manuscript.

Well-written review on Plasma cirRNA as Biomarkers for Breast Cancer .. . There is an unmet need to develop early biomarkers for all cancers in general, BC in particular, due to the severe nature of this disease. The review has done some good graphic representation, and I suppose everything was done under the Biorender /similar tools with proper agreement and license. I have several concerns about the review in its present form. These are summarized below:

  1. The title itself: The portion “ State of art and clinical perspective,” I found hardly there is any discussion of the state-of-the-art tools or technology described in this review. This extra add-on seems unnecessary and largely irrelevant. It may be deleted from the title 

As suggested, “State of the art and clinical perspective” has been removed from the title. The current title is “Plasma Circular RNA as Biomarkers for Breast Cancer”.

  1. The very first section on Page # 1 failed to justify why we need to study cirRNA for BC specifically. Why is the present targets in BC, as well as the existing tools/methodology/technology, lacking, and hence we need to focus on rRNA? BC is perhaps one of the most well-explored cancers, with thousands of sequencing data on mRNA, exome, mutations, etc, well documented. How our focus on cirRNA may add value to this existing knowledge.

According to the request, we have added further considerations in the “Breast cancer” section to better justify the benefits of the circRNA study in BC (lines 43-50):

In recent years, the use of liquid biopsies has become particularly interesting for the detection of circular RNAs (circRNAs), a class of non-coding RNAs (ncRNAs) that are attracting increasing attention as modulators of gene expression through interaction with DNA, miRNAs, and proteins. The findings on the function of circRNAs point to a new hierarchical level in biological process regulation, and emphasise the need for their in-depth investigation. Indeed, dysregulation of circRNA expression is often associated with the on-set and progression of various human diseases, including cancer. Accordingly, circRNAs may represent novel and effective diagnostic, prognostic, and predictive biomarkers for human diseases. In addition, the higher stability compared to linear RNAs makes circRNAs excellent candidates for non-invasive biomarkers in liquid biopsy. In the following sections, we will discuss the most interesting evidence for the potential role of circulating circRNAs (ccircRNA) in plasma as BC biomarkers.”

  1. Section 3, Page 6 ( Liquid Biopsy) section needs to be reframed and justified. The first line “ Liquid biopsy refers to a class of biological fluids of great clinical relevance..” . seems incorrect as liquid biopsy is not a class of biological fluids. it’s a technique. 

As suggested, we have reworded the first sentence of the section “Liquid biopsy and circulating circRNAs” (lines 200-201):

Liquid biopsy is the analysis of molecules from biological fluids of great clinical relevance, such as blood, urine, cerebrospinal fluid, ascites and pleural fluid 

  1. Why do the authors just want to focus on liquid biopsy as the only means for detecting cirRNA? cirRNA can also be assayed in cellular material? Please elaborate on biological samples where cirRNA can be assayed

As mentioned in the section “BC-associated ccircRNAs”, circRNAs can be collected and analysed from tissues and cell lines. However, we wanted to focus on liquid biopsy of plasma as this is a minimally invasive technique to obtain cell-free nucleic acids such as circRNAs. The latter also exhibit greater stability than linear RNAs and overcome the large fragmentation observed in cell-free circulating nucleic acids. Accordingly, circRNAs from plasma may represent promising new biomarkers for the clinical management of BC.

  1. I was expecting a section where the authors discuss the bioinformatics aspect of detecting cirRNA as this is a challenging task and worth discussing. 

To the best of our knowledge, we have made some considerations on the bioinformatic aspect of circRNA identification in the "Conclusions" section (lines 695-704):

Although knowledge about the role of circRNAs in BC is still in its infancy, there are many indications that they could be useful as biomarkers for the diagnosis and prognosis of this disease. The identification of circRNAs from an RNA-seq dataset using various algorithms and bioinformatics tools is an important and challenging step in the study of circRNA. Despite the development of several pipelines for the bioinformatic detection of circRNAs, some problems still need to be solved. For example, the alignment of short-read fragments could be affected by ambiguous read mapping events, especially for genomic sites with similar sequences. In addition, the lack of a gold standard for RNA library preparation, RNA-seq approach and bioinformatics pipeline hinders the appropriate and reproducible identification of circRNAs on a genome-wide scale [119].

  1. section 4 : BC-associated ccircRNA is very long and needs to be fragmented into 4-5 subsections like : maybe cell line-based reporting vs primary cancer-based reporting or cancer stage-specific ccirRNA expression, the role of cirRNA in cancer immune evasion, drug resistance. Rather than one long description, it will be helpful to break in subtopics as discussed

As suggested, we have divided the section “BC-associated ccircRNAs” into 4 subsections:

4.1 Identification of ccircRNAs by expression profiling of plasma (Line 256)

4.2 Identification of ccircRNAs by expression profiling of tissue (Line 333)

4.3 Identification of ccircRNAs by literature and database screening (Line 414)

4.4 TNBC-associated ccircRNAs

  1. A section on diagnostic challenges and their remedy in detecting cirRNA needs to be added 

As discussed in the “Conclusions” section, circRNAs from plasma circRNAs show great diagnostic potential, but many questions still need to be addressed for their employment in routine clinical routine. For example, several circRNAs show a discordance in expression between tissues and plasma, which is probably due to specific and different secretion modes in the bloodstream. Most cancer-associated circRNAs in plasma show no diagnostic value in more than one study, limiting their potential as reliable biomarkers. Furthermore, oncosuppressive circRNAs are generally expressed at low levels in plasma and are therefore barely detectable, leading to their clinical utility being underestimated. Therefore, a deep and robust knowledge of the biological role, mode of secretion in blood and clinical value of cancer-associated ccircRNAs needs to be gathered to determine their clinical utility.

  1. A graphical summary explaining the pleiotropic role of cirRNA in BC is recommended 

As recommended, we have added a box in the new Figure 4 listing has_circ_0069094 as the only circRNA discussed in the review with a proven pleiotropic role in BC.

Round 2

Reviewer 1 Report

Comments and Suggestions for Authors

The revised manuscript has made a great improvement. I have no more comments and recommends.

Reviewer 2 Report

Comments and Suggestions for Authors

Addressed the queries and modified the manuscript. Maybe accepted